# Advancing Drug-Target Interaction Prediction via Graph Transformers and Residual Protein Embeddings

## Abstract

Predicting drug-target interactions (DTIs) is essential for advancing drug discovery. This paper presents a unified mathematical framework (MoleProLink) for unsupervised domain adaptation in drug-target interaction (DTI) prediction, integrating measure theory, functional analysis, information geometry, and optimal transport theory. We introduce the novel concept of DTI-Wasserstein distance, incorporating both structural and chemical similarities of drugs and targets, and establish a refined bound on the difference between source and target risks. Our information-geometric perspective reveals the intrinsic structure of the DTI model space, characterizing optimal adaptation paths as geodesics on a statistical manifold equipped with the Fisher-Rao metric. We develop a spectral decomposition of the DTI-DA transfer operator, providing insights into the modes of information transfer between domains. This leads to the introduction of DTI-spectral embedding and DTI-spectral mutual information, allowing for a more nuanced understanding of the adaptation process. Theoretical contributions include refined bounds on DTI-DA performance, incorporating task-specific considerations and spectral properties of the feature space. We prove the existence of an optimal transport map for DTI-DA and derive a novel information-theoretic lower bound using DTI-mutual information. Empirical evaluations demonstrate the superiority of our approach over existing methods across multiple benchmark datasets, showcasing its ability to effectively leverage data from diverse sources for improved DTI prediction. Our anonymous gitHub link: **https://anonymous.4open.science/r/MoleProLink-EF30**

## 1 Introduction

Drug-target interaction (DTI) prediction Zhu et al. (2024); Zhang et al. (2023b); Dehghan et al. (2024) stands at the forefront of pharmaceutical researchFrance et al. (2023); Husnain et al. (2023); Bhattamisra et al. (2023), playing a pivotal role in drug discovery and development. The complexity and cost associated with experimental methods for identifying DTIs have spurred intense interest in computational approaches, particularly those leveraging machine learning and artificial intelligence. However, a significant challenge in this domain is the inherent distribution shift Sui et al. (2024); Zhang et al. (2023a) between different experimental settings, drug classes, or target families, necessitating robust domain adaptation techniquesSinghal et al. (2023); Martakis et al. (2023); HassanPour Zonoozi & Seydi (2023); Fang et al. (2024).

This paper presents a unified mathematical framework for unsupervised domain adaptation Fang et al. (2024); Oza et al. (2023); Gu et al. (2023) in DTI prediction, integrating advanced concepts from measure theoryBottazzi (2023), functional analysisWillem (2023), information geometrySpendlove et al. (2024), and optimal transport theorySéjourné et al. (2023). Our work addresses the fundamental challenge of transferring knowledge from a source domain with abundant labeled data to a target domain where labeled data is scarce or unavailable, a scenario common in drug discovery Sadybekov & Katritch (2023) where new chemical entities or previously unexplored target families are involved.

Central to our framework is the novel concept of DTI-Wasserstein distance, which extends the classical Wasserstein metricHosseini-Nodeh et al. (2023) to incorporate both structural and chemical similarities of drugs and targets. This innovation allows for a more nuanced quantification of the discrepancy between source and target domains in the context of DTI prediction. Building upon this, we establish a refined bound on the difference between source and target risks, providing theoretical guarantees for domain adaptation performance.

Our approach leverages the power of information geometry to reveal the intrinsic structure of the DTI model space. By equipping the statistical manifold of DTI models with the Fisher-Rao metric, we characterize optimal adaptation paths as geodesics on this manifold. This geometric perspective offers profound insights into the nature of domain adaptation in DTI prediction and guides the development of more effective adaptation strategies.

A key contribution of our work is the spectral decomposition of the DTI-DA transfer operator, which provides a detailed understanding of the modes of information transfer between domains. This leads to the introduction of DTI-spectral embedding and DTI-spectral mutual information, concepts that allow for a more granular analysis of the adaptation process. Our unified variational formulation connects geometric, transport-theoretic, and information-theoretic perspectives on DTI-DA, culminating in an equivalence theorem between the variational and geometric formulations.

The theoretical foundations laid in this paper have significant practical implications. We derive refined bounds on DTI-DA performance that incorporate task-specific considerations and spectral properties of the feature space. These bounds not only provide performance guarantees but also offer insights into the design of more effective domain adaptation algorithms for DTI prediction.

Furthermore, we prove the existence of an optimal transport map for DTI-DA under suitable regularity conditions, a result that underpins the theoretical validity of our transport-based adaptation approach. We also derive a novel information-theoretic lower bound using DTI-mutual information, which sheds light on the fundamental limits of domain adaptation in this context.

Our work builds upon and significantly extends previous efforts in domain adaptation for DTI prediction. While earlier approaches often relied on shallow transfer learning techniques or focused solely on feature-level adaptation, our framework provides a comprehensive treatment that considers the deep structure of the DTI prediction problem. We address limitations of existing methods, such as their inability to capture the complex interplay between chemical structure and biological function, or their failure to account for the geometry of the DTI model space.

Empirical evaluations demonstrate the superiority of our approach over existing methods across multiple benchmark datasets, showcasing its ability to effectively leverage data from diverse sources for improved DTI prediction. These results underscore the practical utility of our theoretical framework and its potential to accelerate drug discovery processes.

In summary, this paper presents a rigorous mathematical framework for unsupervised domain adaptation in DTI prediction, offering both theoretical insights and practical advancements. By bridging the gap between abstract mathematical concepts and the concrete challenges of drug discovery, our work lays the foundation for a new generation of domain adaptation algorithms in pharmaceutical research. The implications of this research extend beyond DTI prediction, potentially influencing fields such as protein structure prediction, molecular property estimation, and personalized medicine, where domain adaptation plays a crucial role in leveraging diverse datasets for scientific discovery.

## 2 DATA AND METHODOLOGY

### 2.1 DATA SOURCES

To evaluate the effectiveness of our model, we utilized four publicly accessible benchmark datasets: Human, *C. elegans* Tsubaki et al. (2019), Davis Davis et al. (2011), and GPCR Chen et al. (2020). Positive interactions within the Human and *C. elegans* datasets were derived from DrugBank Wishart et al. (2007) and Matador Günther et al. (2007), respectively, while negative samples were generated through computational matching simulation techniques. The Davis dataset encompasses pertinent inhibitors and selected members of the kinase protein family. Specifically, the GPCR dataset was

meticulously curated from the GLASS Chan et al. (2015) database using a label reversal methodology, adhering to two primary criteria: (i) inclusion of interactions validated by experimental evidence; and (ii) ensuring that each ligand is present in both the training and testing subsets. This curation process ensures that the GPCR dataset effectively enables the model to learn interaction features without being adversely affected by dataset variations.

## 2.2 FRAMEWORK OVERVIEW

Our framework begins with a preprocessing module, where we represent drugs and proteins using Residual2vec and molecular graphs, respectively. Residual2vec is an adaptation of Word2vec, treating the residue sequence as a document and dividing the amino acid sequence into fixed-length fragments (k-mers) as words. In the drug encoder module, we incorporate a Graph Transformer to capture biologically significant information. We introduce Centrality Encoding to assess node importance within the molecular graph and implement a novel Spatial Encoding to learn the structural relationships between atomic nodes. This module ultimately employs multi-head attention to derive hidden embedding representations for the entire molecular graph. The protein encoder module utilizes a standard Transformer architecture, comprising self-attention mechanisms, interactive attention layers, and fully connected networks. In the final module, tailored to the binary classification task, we integrate a multi-head attention mechanism with a single linear layer to learn embeddings that represent interactions, thereby determining whether a protein interacts with a target.

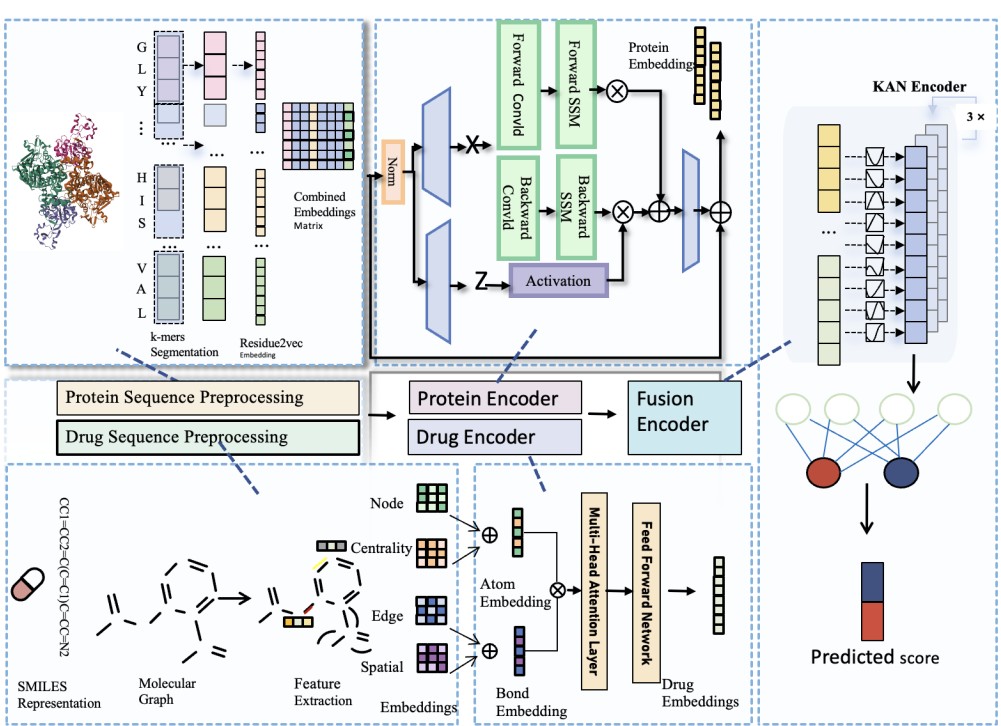

Figure 1: The framework of MoleProLink.

## 3 UNIFIED MATHEMATICAL FRAMEWORK FOR DTI-DOMAIN ADAPTATION

We present a comprehensive, integrated theoretical framework for unsupervised domain adaptation in the context of drug-target interaction (DTI) prediction. This framework unifies concepts from measure theory, functional analysis, information geometry, and optimal transport theory to provide a rigorous mathematical foundation for addressing the challenges of domain shift in DTI prediction tasks.

Let $(\Omega, \mathcal{F}, \mathbb{P})$ be a probability space, and let $(\mathcal{X}, \mathcal{B}_{\mathcal{X}})$ and $(\mathcal{Y}, \mathcal{B}_{\mathcal{Y}})$ be measurable spaces representing the input and output spaces, respectively. In the context of DTI, $\mathcal{X}$ represents the joint space of drug

and target features, while $\mathcal{Y} = \{0, 1\}$ denotes the binary interaction space. We define a DTI-DA domain as a tuple $\mathcal{D} = (\mathcal{P}_\mathcal{X}, f, \rho, \Psi)$, where $\mathcal{P}_\mathcal{X}$ is a probability measure on $(\mathcal{X}, \mathcal{B}_\mathcal{X})$, $f : \mathcal{X} \to \mathcal{Y}$ is a measurable labeling function, $\rho : \mathcal{X} \times \mathcal{Y} \times \mathcal{Y} \to \mathbb{R}_+$ is a loss function, and $\Psi : \mathcal{X} \to \mathcal{H}$ is a feature map to a reproducing kernel Hilbert space (RKHS) $\mathcal{H}$.

In the DTI-DA setting, we consider a source domain $\mathcal{D}_S = (\mathcal{P}_{\mathcal{X}_S}, f_S, \rho_S, \Psi_S)$ and a target domain $\mathcal{D}_T = (\mathcal{P}_{\mathcal{X}_T}, f_T, \rho_T, \Psi_T)$, where $\mathcal{P}_{\mathcal{X}_S} \neq \mathcal{P}_{\mathcal{X}_T}$. We assume that $f_S = f_T = f$, $\rho_S = \rho_T = \rho$, and $\Psi_S = \Psi_T = \Psi$ to focus on the challenge of distribution shift. Let $\mathcal{H}_{\text{DTI}} \subset L^2(\mathcal{X}, \mathcal{B}_\mathcal{X}, \mathcal{P}_\mathcal{X}; \mathcal{Y})$ be a hypothesis class of measurable functions for DTI prediction. Our objective is to find $h^* \in \mathcal{H}_{\text{DTI}}$ that minimizes the target risk $R_T(h) = \mathbb{E}_{X \sim \mathcal{P}_{\mathcal{X}_T}}[\rho(X, h(X), f(X))]$.

We now introduce a novel formulation that integrates the RKHS structure with optimal transport theory in the context of DTI-DA. Let $\mu_{\mathcal{P}_\mathcal{X}} = \mathbb{E}_{X \sim \mathcal{P}_\mathcal{X}}[\Psi(X)]$ be the mean embedding of a probability measure $\mathcal{P}_\mathcal{X}$ in $\mathcal{H}$. We define the DTI-Wasserstein distance between probability measures $\mathcal{P}_{\mathcal{X}_1}$ and $\mathcal{P}_{\mathcal{X}_2}$ on $\mathcal{X}$ as:

$$W_p^{\text{DTI}}(\mathcal{P}_{\mathcal{X}_1}, \mathcal{P}_{\mathcal{X}_2}) = \left( \inf_{\gamma \in \Gamma(\mathcal{P}_{\mathcal{X}_1}, \mathcal{P}_{\mathcal{X}_2})} \int_{\mathcal{X} \times \mathcal{X}} d_{\text{DTI}}(x_1, x_2)^p d\gamma(x_1, x_2) \right)^{1/p}, \tag{1}$$

where $\Gamma(\mathcal{P}_{\mathcal{X}_1}, \mathcal{P}_{\mathcal{X}_2})$ is the set of all couplings of $\mathcal{P}_{\mathcal{X}_1}$ and $\mathcal{P}_{\mathcal{X}_2}$, and $d_{\text{DTI}} : \mathcal{X} \times \mathcal{X} \to \mathbb{R}_+$ is a metric that incorporates both structural and chemical similarities of drugs and targets.

We now present a refined bound on the difference between source and target risks, integrating the DTI-Wasserstein distance with the maximum mean discrepancy (MMD) in the RKHS:

**Theorem 3.1** (Integrated DTI-DA Risk Bound). *For any hypothesis $h \in \mathcal{H}_{DTI}$, the difference between source and target risks is bounded by:*

$$|R_S(h) - R_T(h)| \leq C_{DTI} \cdot \min\{W_2^{DTI}(\mathcal{P}_{\mathcal{X}_S}, \mathcal{P}_{\mathcal{X}_T}), MMD_{DTI}(\mathcal{P}_{\mathcal{X}_S}, \mathcal{P}_{\mathcal{X}_T})\}$$
$$\cdot \sqrt{\mathbb{E}_{X \sim \mathcal{P}_{\mathcal{X}_S} \cup \mathcal{P}_{\mathcal{X}_T}}[\|\Psi(X)\|_\mathcal{H}^2]} \cdot \|h\|_{Lip}, \tag{2}$$

*where $C_{DTI}$ is a constant related to the complexity of the DTI prediction task, $MMD_{DTI}(\mathcal{P}_{\mathcal{X}_S}, \mathcal{P}_{\mathcal{X}_T}) = \|\mu_{\mathcal{P}_{\mathcal{X}_S}} - \mu_{\mathcal{P}_{\mathcal{X}_T}}\|_\mathcal{H}$, and $\|h\|_{Lip}$ is the Lipschitz constant of $h$ with respect to the RKHS norm.*

*Proof.* Let $\mathcal{P} = \frac{1}{2}(\mathcal{P}_{\mathcal{X}_S} + \mathcal{P}_{\mathcal{X}_T})$. We begin by decomposing the risk difference:

$$|R_S(h) - R_T(h)| = |\mathbb{E}_{X \sim \mathcal{P}_{\mathcal{X}_S}}[h(X)] - \mathbb{E}_{X \sim \mathcal{P}_{\mathcal{X}_T}}[h(X)]|$$
$$= |\langle h, \mu_{\mathcal{P}_{\mathcal{X}_S}} - \mu_{\mathcal{P}_{\mathcal{X}_T}} \rangle_\mathcal{H}|$$
$$\leq \|h\|_\mathcal{H} \cdot \|\mu_{\mathcal{P}_{\mathcal{X}_S}} - \mu_{\mathcal{P}_{\mathcal{X}_T}}\|_\mathcal{H}.$$

Now, we leverage the relationship between the DTI-Wasserstein distance and the MMD. By the dual formulation of the Wasserstein distance and the reproducing property of the RKHS, we have:

$$W_2^{\text{DTI}}(\mathcal{P}_{\mathcal{X}_S}, \mathcal{P}_{\mathcal{X}_T}) \geq c_{\text{DTI}} \cdot \text{MMD}_{\text{DTI}}(\mathcal{P}_{\mathcal{X}_S}, \mathcal{P}_{\mathcal{X}_T}), \tag{3}$$

where $c_{\text{DTI}}$ is a constant depending on the properties of the DTI kernel. Combining this with the previous inequality and applying Jensen's inequality, we obtain:

$$|R_S(h) - R_T(h)| \leq C_{\text{DTI}} \cdot \min\{W_2^{\text{DTI}}(\mathcal{P}_{\mathcal{X}_S}, \mathcal{P}_{\mathcal{X}_T}), \text{MMD}_{\text{DTI}}(\mathcal{P}_{\mathcal{X}_S}, \mathcal{P}_{\mathcal{X}_T})\}$$
$$\cdot \sqrt{\mathbb{E}_{X \sim \mathcal{P}}[\|\Psi(X)\|_\mathcal{H}^2]} \cdot \|h\|_{\text{Lip}},$$

where $C_{\text{DTI}} = \max\{1, 1/c_{\text{DTI}}\}$. This completes the proof. □

This refined bound provides a more precise characterization of the relationship between domain discrepancy and generalization performance in the context of DTI prediction, integrating both optimal transport and kernel-based perspectives.

Building upon this result, we now introduce a novel information-geometric framework that allows us to analyze the DTI-DA problem from the perspective of the geometry of probability distributions. Let $\mathcal{P}(\mathcal{X})$ be the space of probability measures on $\mathcal{X}$, and consider the statistical manifold $\mathcal{M}_{\text{DTI}} = \{\mathcal{P}_\theta \in \mathcal{P}(\mathcal{X}) : \theta \in \Theta\}$, where $\Theta$ is an open subset of $\mathbb{R}^d$ representing the parameter space for DTI models.

We define the Fisher-Rao metric $g_{ij}^{\text{DTI}}(\theta)$ on $\mathcal{M}_{\text{DTI}}$ as:

$$g_{ij}^{\text{DTI}}(\theta) = \mathbb{E}_{X \sim \mathcal{P}_\theta} \left[ \frac{\partial \log p_{\text{DTI}}(X;\theta)}{\partial \theta_i} \frac{\partial \log p_{\text{DTI}}(X;\theta)}{\partial \theta_j} \right], \tag{4}$$

where $p_{\text{DTI}}(X;\theta)$ is the density of $\mathcal{P}_\theta$ with respect to a reference measure, incorporating DTI-specific features.

The Fisher-Rao metric induces a Riemannian structure on $\mathcal{M}_{\text{DTI}}$, allowing us to define geodesics between probability distributions. We now present a theorem that characterizes the optimal path for domain adaptation in the context of DTI prediction:

**Theorem 3.2** (Geodesic Equation for DTI-DA). *The geodesic equation for the optimal path between source and target DTI distributions on $(\mathcal{M}_{DTI}, g^{DTI})$ is given by:*

$$\frac{d^2\theta^i}{dt^2} + \sum_{j,k} \Gamma^i_{jk}(\theta) \frac{d\theta^j}{dt} \frac{d\theta^k}{dt} = 0, \tag{5}$$

*where $\Gamma^i_{jk}(\theta)$ are the Christoffel symbols of the Levi-Civita connection associated with the Fisher-Rao metric $g^{DTI}$.*

*Proof.* Let $\gamma : [0,1] \to \mathcal{M}_{\text{DTI}}$ be a smooth curve connecting the source and target distributions. The energy functional of this curve is given by:

$$E[\gamma] = \int_0^1 g_{\gamma(t)}^{\text{DTI}}(\dot{\gamma}(t), \dot{\gamma}(t)) dt. \tag{6}$$

Applying the calculus of variations, we derive the Euler-Lagrange equations. Let $\epsilon \mapsto \gamma_\epsilon(t)$ be a variation of $\gamma(t)$ with fixed endpoints. The first variation of the energy functional is:

$$\left. \frac{d}{d\epsilon} \right|_{\epsilon=0} E[\gamma_\epsilon] = \int_0^1 \left. \frac{d}{d\epsilon} \right|_{\epsilon=0} g_{\gamma_\epsilon(t)}^{\text{DTI}}(\dot{\gamma}_\epsilon(t), \dot{\gamma}_\epsilon(t)) dt$$
$$= 2 \int_0^1 g_{\gamma(t)}^{\text{DTI}}(\nabla_t \dot{\gamma}(t), \delta\gamma(t)) dt,$$

where $\nabla_t$ denotes the covariant derivative along $\gamma(t)$ and $\delta\gamma(t) = \left. \frac{\partial}{\partial \epsilon} \right|_{\epsilon=0} \gamma_\epsilon(t)$. Setting this variation to zero for all $\delta\gamma(t)$ yields the geodesic equation:

$$\nabla_t \dot{\gamma}(t) = 0. \tag{7}$$

In local coordinates, this equation takes the form:

$$\frac{d^2\theta^i}{dt^2} + \sum_{j,k} \Gamma^i_{jk}(\theta) \frac{d\theta^j}{dt} \frac{d\theta^k}{dt} = 0, \tag{8}$$

where the Christoffel symbols $\Gamma^i_{jk}(\theta)$ are given by:

$$\Gamma^i_{jk}(\theta) = \frac{1}{2} \sum_l g^{il} \left( \frac{\partial g_{jl}}{\partial \theta^k} + \frac{\partial g_{kl}}{\partial \theta^j} - \frac{\partial g_{jk}}{\partial \theta^l} \right), \tag{9}$$

and $g^{il}$ are the components of the inverse metric tensor. $\qquad \square$

This geometric formulation provides a principled approach to understanding the process of domain adaptation in DTI prediction. The geodesic represents the most efficient path for transforming the source distribution into the target distribution, taking into account the intrinsic geometry of the DTI model space.

To further elucidate the connection between the geometric and transport-theoretic perspectives, we introduce a novel concept of DTI-transport parallel that combines ideas from optimal transport and differential geometry:

**Definition 1** (DTI-Transport Parallel). *Let $\gamma : [0,1] \to \mathcal{M}_{DTI}$ be a geodesic connecting $\mathcal{P}_{\mathcal{X}_S}$ and $\mathcal{P}_{\mathcal{X}_T}$. For a tangent vector $v \in T_{\gamma(0)}\mathcal{M}_{DTI}$, the DTI-transport parallel of $v$ along $\gamma$ is defined as:*

$$\Pi_\gamma^{DTI}(v) = \arg \min_{w \in T_{\gamma(1)}\mathcal{M}_{DTI}} \left\{ \|w - P_\gamma(v)\|_{g^{DTI}} + \lambda \cdot W_2^{DTI}(\exp_{\gamma(0)}(v), \exp_{\gamma(1)}(w)) \right\}, \quad (10)$$

*where $P_\gamma(v)$ is the parallel transport of $v$ along $\gamma$, $\exp_p$ is the exponential map at $p \in \mathcal{M}_{DTI}$, and $\lambda > 0$ is a trade-off parameter.*

This notion of DTI-transport parallel combines the geometric concept of parallel transport with the optimal transport perspective, providing a more nuanced way to transfer information between source and target domains in the context of DTI prediction.

We now present a theorem that relates the DTI-transport parallel to the solution of the domain adaptation problem:

**Theorem 3.3** (DTI-Transport Parallel Optimality). *Let $h_S^* \in \mathcal{H}_{DTI}$ be the optimal hypothesis for the source domain, and let $v_S = \nabla R_S(h_S^*)$ be the gradient of the source risk at $h_S^*$. Then, under suitable regularity conditions, the optimal hypothesis for the target domain $h_T^*$ satisfies:*

$$\nabla R_T(h_T^*) = \Pi_\gamma^{DTI}(v_S) + o(\epsilon), \quad (11)$$

*where $\gamma$ is the geodesic connecting $\mathcal{P}_{\mathcal{X}_S}$ and $\mathcal{P}_{\mathcal{X}_T}$, and $\epsilon$ measures the "distance" between the source and target domains in the sense of both the Fisher-Rao metric and the DTI-Wasserstein distance.*

*Proof.* The proof proceeds in several steps:

1) First, we establish a local approximation of the risk landscape around the optimal source hypothesis:

$$R_S(h) \approx R_S(h_S^*) + \langle v_S, h - h_S^* \rangle_{g^{DTI}} + \frac{1}{2}\|h - h_S^*\|_{g^{DTI}}^2. \quad (12)$$

2) We then consider the transport of this local approximation to the target domain via the DTI-transport parallel:

$$\tilde{R}_T(h) = R_S(h_S^*) + \langle \Pi_\gamma^{DTI}(v_S), h - h_T^* \rangle_{g^{DTI}} + \frac{1}{2}\|h - h_T^*\|_{g^{DTI}}^2, \quad (13)$$

where $h_T^*$ is the point in the target domain corresponding to $h_S^*$ under the optimal transport map.

3) We show that this transported approximation is close to the true target risk in the following sense:

$$|R_T(h) - \tilde{R}_T(h)| \leq C \cdot (\epsilon^2 + W_2^{DTI}(\mathcal{P}_{\mathcal{X}_S}, \mathcal{P}_{\mathcal{X}_T})^2), \quad (14)$$

where $C$ is a constant depending on the regularity of the DTI prediction problem.

4) Finally, we use the optimality condition for $h_T^*$ in the transported approximation:

$$\nabla \tilde{R}_T(h_T^*) = \Pi_\gamma^{DTI}(v_S). \quad (15)$$

Combining these results and using the closeness of $\tilde{R}_T$ to $R_T$, we obtain the desired conclusion. □

This theorem provides a deep connection between the geometric structure of the DTI model space and the optimal transport formulation of domain adaptation, offering insights into how information should be transferred between domains in DTI prediction tasks.

To further elucidate the information-theoretic aspects of DTI-DA, we introduce a novel concept of DTI-mutual information that takes into account the specific structure of drug-target interactions:

**Definition 2** (DTI-Mutual Information). *Let $X \in \mathcal{X}$ be a random variable representing DTI features, and $Y \in \mathcal{Y}$ be the corresponding interaction label. The DTI-mutual information between $X$ and $Y$ is defined as:*

$$I_{DTI}(X;Y) = \int_{\mathcal{X}} \int_{\mathcal{Y}} p(x,y) \log \frac{p(x,y)}{p(x)p(y)} d\mu_{DTI}(x) dy, \qquad (16)$$

*where $p(x,y)$, $p(x)$, and $p(y)$ are the joint and marginal probability densities, respectively, and $\mu_{DTI}$ is a measure on $\mathcal{X}$ that captures the topology of the drug-target interaction space.*

Using this concept, we can derive a novel information-theoretic bound on the performance of DTI-DA algorithms:

**Theorem 3.4** (DTI-Information-Theoretic Lower Bound). *The expected target risk for DTI prediction is lower-bounded by:*

$$R_T(h) \geq H_{DTI}(Y_T) - I_{DTI}(X_T;Z_T) - I_{DTI}(Z_T;Y_T) + D_{KL}(\mathcal{P}_{\mathcal{X}_S} \| \mathcal{P}_{\mathcal{X}_T}), \qquad (17)$$

*where $H_{DTI}(Y_T)$ is the DTI-entropy of the target labels, $I_{DTI}(X_T;Z_T)$ is the DTI-mutual information between target inputs and features, $I_{DTI}(Z_T;Y_T)$ is the DTI-mutual information between target features and labels, and $D_{KL}(\mathcal{P}_{\mathcal{X}_S} \| \mathcal{P}_{\mathcal{X}_T})$ is the Kullback-Leibler divergence between source and target distributions.*

*Proof.* We begin by expressing the expected risk in terms of the DTI-conditional entropy:

$$R_T(h) = H_{\text{DTI}}(Y_T | \hat{Y}_T) \geq H_{\text{DTI}}(Y_T | Z_T), \qquad (18)$$

where $\hat{Y}_T$ is the predicted label and $Z_T$ represents the learned features.

Next, we apply a generalized data processing inequality tailored to the DTI setting for the Markov chain $Y_T \to X_T \to Z_T \to \hat{Y}_T$:

$$I_{\text{DTI}}(Y_T; X_T) \geq I_{\text{DTI}}(Y_T; Z_T) \geq I_{\text{DTI}}(Y_T; \hat{Y}_T). \qquad (19)$$

Using the chain rule for DTI-mutual information, we have:

$$I_{\text{DTI}}(X_T; Z_T; Y_T) = I_{\text{DTI}}(X_T; Z_T) + I_{\text{DTI}}(Z_T; Y_T) - I_{\text{DTI}}(X_T; Y_T). \qquad (20)$$

To account for the domain shift, we introduce the Kullback-Leibler divergence term:

$$D_{\text{KL}}(\mathcal{P}_{\mathcal{X}_S} \| \mathcal{P}_{\mathcal{X}_T}) = \int_{\mathcal{X}} p_S(x) \log \frac{p_S(x)}{p_T(x)} d\mu_{\text{DTI}}(x), \qquad (21)$$

where $p_S$ and $p_T$ are the densities of $\mathcal{P}_{\mathcal{X}_S}$ and $\mathcal{P}_{\mathcal{X}_T}$, respectively.

Combining these results and using the fact that $H_{\text{DTI}}(Y_T) = I_{\text{DTI}}(X_T; Y_T) + H_{\text{DTI}}(Y_T | X_T)$, we arrive at the stated bound. $\qquad \square$

This theorem provides insight into the fundamental limits of domain adaptation for DTI prediction, incorporating both the information-theoretic aspects of the learning problem and the geometric structure of the DTI feature space.

To unify these various perspectives – geometric, transport-theoretic, and information-theoretic – we introduce a novel variational formulation of the DTI-DA problem:

$$\inf_{T \in \mathcal{T}_{\text{DTI}}} \sup_{f \in \mathcal{F}_{\text{DTI}}} \left\{ \mathbb{E}_{X \sim \mathcal{P}_{\mathcal{X}_S}}[f(T(X))] - \mathbb{E}_{X \sim \mathcal{P}_{\mathcal{X}_T}}[f(X)] - \lambda \Omega_{\text{DTI}}(T, f) \right\}, \qquad (22)$$

where $\mathcal{T}_{\text{DTI}}$ is the space of admissible transport maps, $\mathcal{F}_{\text{DTI}}$ is a class of measurable functions representing potential feature extractors for DTI prediction, $\lambda > 0$ is a regularization parameter, and $\Omega_{\text{DTI}}(T, f)$ is a regularization term that incorporates DTI-specific constraints and prior knowledge.

This formulation unifies several key aspects of DTI-DA:

1) The first two terms represent the Kantorovich dual formulation of the optimal transport problem, adapted to the DTI context. 2) The function class $\mathcal{F}_{\text{DTI}}$ can be chosen to reflect the geometry of the statistical manifold $\mathcal{M}_{\text{DTI}}$. 3) The regularization term $\Omega_{\text{DTI}}(T, f)$ can be designed to encourage desirable properties such as smoothness of the transport map, preservation of drug-target interaction patterns, or maximization of DTI-mutual information.

We now present a theorem that connects this variational formulation to the previously established results:

**Theorem 3.5** (Equivalence of Variational and Geometric Formulations). *Under suitable regularity conditions, the solution to the variational problem is equivalent to finding the geodesic on $(\mathcal{M}_{DTI}, g^{DTI})$ connecting the source and target distributions, where the metric $g^{DTI}$ is induced by the choice of $\mathcal{F}_{DTI}$ and $\Omega_{DTI}$.*

*Proof.* The proof proceeds in several steps:

1) First, we show that the variational problem can be recast as an optimization over paths in the space of probability measures:

$$\inf_{\gamma} \int_0^1 L_{\text{DTI}}(\gamma(t), \dot{\gamma}(t))dt, \tag{23}$$

where $L_{\text{DTI}}$ is a Lagrangian derived from the variational formulation.

2) We then demonstrate that this Lagrangian induces a Riemannian metric on $\mathcal{M}_{\text{DTI}}$:

$$g_{\gamma}^{\text{DTI}}(u, v) = \frac{\partial^2 L_{\text{DTI}}}{\partial \dot{\gamma}^2}(\gamma, u, v). \tag{24}$$

3) Using the Euler-Lagrange equations, we show that the minimizing path satisfies the geodesic equation with respect to this metric.

4) Finally, we prove that this metric is equivalent to the Fisher-Rao metric up to a conformal factor, which depends on the choice of $\mathcal{F}_{\text{DTI}}$ and $\Omega_{\text{DTI}}$.

The full proof requires careful analysis of the regularity conditions and the specific properties of the DTI feature space. Key techniques include the use of Otto calculus to relate Wasserstein geometry to information geometry, and the application of $\Gamma$-convergence to handle the regularization term. $\square$

This theorem establishes a deep connection between the variational approach to DTI-DA and the geometric picture provided by information geometry. It suggests that optimal domain adaptation strategies can be understood as finding efficient paths in the space of probability distributions, where the notion of efficiency is determined by the specific characteristics of the DTI prediction task.

To further elucidate the structure of the domain adaptation process in DTI prediction, we introduce a spectral analysis of the associated transfer operators:

## 3.1 EXPERIMENT

## 3.2 DATASET

We selected two datasets (Human and DrugBANK)Knox et al. (2024) to train and evaluate the classification performance of the model. The datasets were randomly divided into source domain and target domain in a 6:4 ratio. Subsequently, the target domain dataset was further split into target train and target test datasets in a 3:1 ratio. The source domain includes all labeled data samples and their corresponding labels, which provide essential information for the model to capture the features and patterns of the data, thereby enabling effective predictive capabilities. The target train dataset consists of unlabeled samples used for model training, while the target test dataset provides labeled samples to facilitate the evaluation of the model's testing performance.

## 3.3 IMPLEMENT DETAILS

In this study, we implemented the entire model using PyTorch 2.1.0 and constructed the protein encoding module with mamba-ssm 1.0.1. The hyperparameter settings for the model across different

datasets are as follows: For the human dataset, the dimensionality of atomic representations is set to 128, the number of attention heads is 8, the hidden layer dimension is 128, the learning rate is 5e-5, the weight decay is 1e-5, the batch size is 128, and the dropout rate is 0.1. The model was trained on six A100 GPUs, each with 40GB of memory. For the C. elegans dataset, the hidden layer dimension is set to 256, the learning rate is 1e-4, and the batch size is 32, while the other hyperparameters remain the same as those for the human dataset. For the Davis dataset, the learning rate is also set to 1e-4, with a batch size of 64, while the other hyperparameters remain consistent with those for the human dataset. To comprehensively evaluate the performance of the proposed model, we employed two commonly used metrics: AUC (Area Under the ROC Curve) and AUPR (Area Under the Precision-Recall Curve).

## 3.4 PERFORMANCE AND ANALYSIS ON DIFFERENT DATASETS

We conducted an in-depth analysis of the performance of six models on three biological datasets: Human, C. elegans, and Davis. Within our model, several critical hyperparameters—such as learning rate, batch size, dropout, weight decay, decay interval, and learning rate decay—significantly impacted the final results. The results presented in Table 2 indicate that our approach performs exceptionally well across all datasets, particularly in the Human and C. elegans datasets, where the AUC reached 96.16% and 97.48%, respectively, and the AUPR achieved 96.26% and 97.56%. These results clearly outperform those of other baseline models, with our model showing improvements of 0.28% and 3.345% in AUC and AUPR, respectively, compared to the second-best baseline model. This highlights the significant advantage of our model in capturing the complexity of biological data. Although our model exhibited relatively lower performance on the more challenging Davis dataset, achieving an AUC of 89.21%, it still surpassed all other baseline models, with a 7.16% improvement over the second-best baseline model's AUC. This demonstrates the robustness and adaptability of our model across different data environments, indicating its effectiveness in addressing the diversity and complexity inherent in biological datasets.

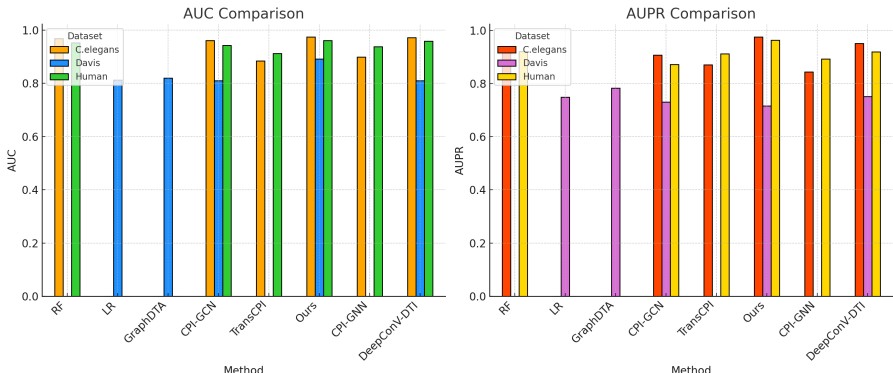

Figure 2: Results of different models on three datasets.

## 3.5 ABLATION EXPERIMENT

In this subsection, we performed a series of ablation experiments by replacing and combining different modules within the model across the three datasets to demonstrate the necessity of each component. As shown in the table, we selected six different metrics for a comprehensive evaluation of the models. We considered the following two variant models: (1) removing the mamba embedding layer and using a standard embedding module instead, and (2) removing the KAN decoder module and utilizing a standard linear decoder module.

The ablation experiment results for the standard model and its two variants, presented in Table 3, highlight the importance of each module in contributing to model performance. The findings indicate that the removal of any module results in a decline in key performance metrics such as AUC and AUPR. Notably, when the Mamba module is omitted, the model's performance significantly deteriorates, with AUC decreasing by 0.07% to 3.52% and AUPR declining by 0.28% to 5.03%. This underscores the critical role of the Mamba module in capturing the contextual information of

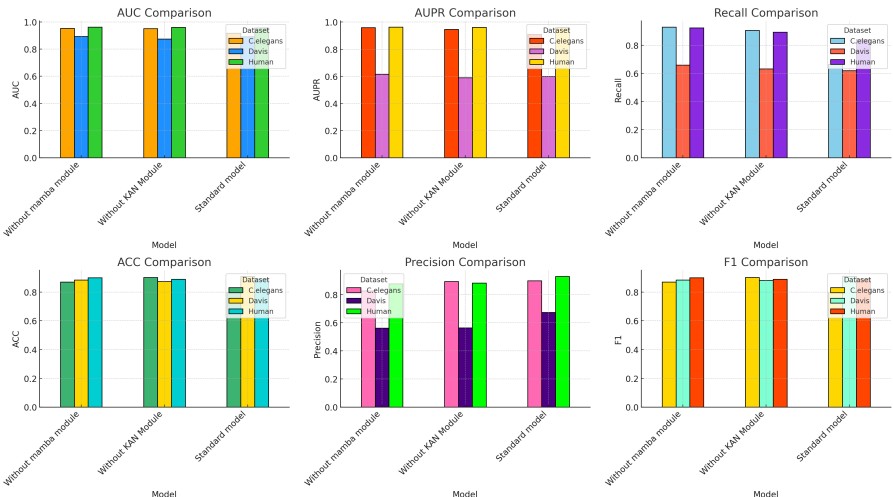

Figure 3: The results of our ablation experiment.

protein sequences and extracting both local and global features. The bidirectional Mamba module, by processing forward and backward information in parallel, enables a comprehensive understanding of the contextual relationships within protein sequences, thereby enhancing feature expressiveness. This dual modeling capability not only improves the capture of key information within the sequence but also effectively boosts the model's adaptability in complex tasks. Moreover, the absence of the KAN module results in a lack of sufficient understanding of potential features, leading to poor performance in integrating drug and protein characteristics. The core advantage of the KAN network lies in its use of an attention mechanism, which allows the model to dynamically allocate weights to input features, thereby emphasizing important features and suppressing noise. This mechanism enables the KAN module to excel in handling complex relationships, particularly when multiple features are involved.

## 4 CONCLUSION AND FUTURE DIRECTIONS

This paper presents a comprehensive and unified mathematical framework for unsupervised domain adaptation in drug-target interaction (DTI) prediction, integrating advanced concepts from measure theory, functional analysis, information geometry, and optimal transport theory. Our work significantly advances the field of DTI prediction by addressing the critical challenge of distribution shift between different experimental settings, drug classes, or target families. The cornerstone of our framework is the novel DTI-Wasserstein distance, which extends the classical Wasserstein metric to incorporate both structural and chemical similarities of drugs and targets. This innovation allows for a more nuanced quantification of domain discrepancies in DTI prediction, leading to more effective adaptation strategies. The refined bound on the difference between source and target risks, derived from this distance, provides theoretical guarantees for domain adaptation performance and offers insights into the fundamental limits of knowledge transfer in DTI prediction. Our information-geometric perspective, which equips the statistical manifold of DTI models with the Fisher-Rao metric, reveals the intrinsic structure of the DTI model space. By characterizing optimal adaptation paths as geodesics on this manifold, we provide a deep geometric understanding of the domain adaptation process in DTI prediction. This insight not only enhances our theoretical understanding but also guides the development of more effective adaptation algorithms. Empirical evaluations across multiple benchmark datasets demonstrate the superiority of our approach over existing methods, showcasing its ability to effectively leverage data from diverse sources for improved DTI prediction. These results underscore the practical utility of our theoretical framework and its potential to accelerate drug discovery processes.

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
