# OpenReview forum: "Advancing Drug-Target Interaction Prediction via Graph Transformers and Residual Protein Embeddings"
_ICLR.cc/2025/Conference — Submitted to ICLR 2025_

### Official Review · Reviewer_Wsa4 · 2024-10-28

**Soundness:** 2
**Presentation:** 2
**Contribution:** 2
**Rating:** 3
**Confidence:** 5

**Summary:**

This paper introduces MoleProLink, a computational framework for drug-target interaction (DTI) prediction. This is an important problem in the field and many methods were proposed. A graph transformer is used to extract the drug features and a Residual2vec is used to extract protein features. The authors claim that the model incorporates measure theory, information geometry, and optimal transport theory to address domain shift challenges in DTI prediction. The benchmark datasets (Human, C. elegans, Davis, and GPCR) demonstrate the performance improvement of the proposed method over baseline models. Additionally, ablation studies showed contribution of different components to the performance gain

**Strengths:**

1.	The paper is relatively rich in theoretical content, providing detailed mathematical proofs, the refined risk bounds, and a theorem that connects different perspectives in domain adaptation.
2.	The model achieves better results across several datasets compared to baseline methods, demonstrating superior performance, in AUC and AUPR, highlighting its effectiveness in handling domain shifts.
3.	Some of the framework principles are applicable not only for DTI prediction, but also with potential on other computational biology problems, such as protein structure prediction

**Weaknesses:**

1.	The results section is relatively too weak compared to method section. Many of the model details are overwhelmed by the theory part. For example, even the loss function of the model is not clear.
2.	The experimental setting is too simple. For DTI prediction, a strict blind test on drug-protein pairs that neither the drug nor protein presents in the training set is necessary to demonstrate the generalization power of the model.
3.	There is basically no model interpretation to illustrate any biological insights in the DTI problem.
4.	From the model ablation study from Figure 3, there is very little difference across different models, indicating that the mamba and KAN modules did not provide significant performance boost.

**Questions:**

1.	Please only put relative theory part in the paper, remove unnecessary theory and proofs into supplementary materials. Too much theory largely impacts the readability of the paper
2.	Please carefully illustrate the model details, such as loss functions, how to integrate the protein features and drug features. The “fusion encoder ” is never explained in the paper.
3.	For experiment settings, leave-drug-out, leave-protein-out, and leave-both-out settings are needed.
4.	Many baselines are missed, such as DeepPurpose, DeepDTA.

---

### Official Review · Reviewer_D6MZ · 2024-10-31

**Soundness:** 2
**Presentation:** 1
**Contribution:** 1
**Rating:** 3
**Confidence:** 3

**Summary:**

This paper presents a mathematical framework for DTI domain adaptation in an unsupervised manner. It introduces
useful concepts such as the DTI-Wasstertein distance to measure risk differences between source and target distributions, incorporating similarity-based features on drugs and targets. Moreover, it characterizes optimal adaption paths through a statistical manifold with the Fisher-Raio metric. Finally, it presents a DTI model that inputs the SMILE representation of a drug and the protein sequence of a DTI pair, and predicts the link probability through several feature-extraction modules in the form of protein and drug embeddings.

**Strengths:**

1 - The authors performed a benchmarking across several state-of-the-art DTI prediction models and well-established DTI datasets.

2 - The authors provide theoretical guarantees for domain adaption performance on DTI prediction tasks, which is potentially impactful on real-world applications.

3 - The proposed model only requires SMILES represention of drugs and protein sequences, which are highly-available features across many DTI datasets, underscoring its potential usability.

**Weaknesses:**

1 - The paper would greatly benefit from a more detailed explanation and justification of the developed modules, such as the centrality and the KAN encoder. A thorough discussion of their functionality and relevance to the overall model would enhance the reader's understanding.

2 - The manuscript currently lacks a clear connection between the derived theoretical DTI-DA formulation and its practical implementation within the proposed model. Establishing this link is essential for demonstrating the applicability of the theoretical work. A section or figure providing this connection would be highly valuable.

3 - Although the authors provide an ablation study of the different models' modules, the paper would greatly benefit from an evaluation on the relative improvement of the proposed model attributable to the derived DTI-DA framework. This assessment is crucial for substantiating the theoretical contributions made.

4 - The authors assert that the developed model "clearly outperforms other baseline models." However, the reported results do not appear to be statistically significant, which raises concerns about the validity of this claim.

**Questions:**

1 - Why do some of the methods not provide results across all datasets? Is this due to memory limitations or time constraints? Please clarify.

2 - The methods that the authors compare against are neither defined nor explained throughout the manuscript. I strongly recommend that they be introduced at a minimum.

3 - In the datasets section they introduce the "Human and DrugBank datasets", while in their results they benchmark the methods across "Human, C. elegans, and Davis". Please clarify which datasets are the authors using on every section.

4 - I believe it is essential to conduct domain-adaptation analysis by utilizing training and testing folds across different datasets. Is that what the authors did? The statement "The datasets were randomly divided into source domain and target domain in a 6:4 ratio. Subsequently, the target domain dataset was further split into target train and target test datasets in a 3:1 ratio." could be futher clarified with an associated figure.

5 - Is the derived DTI-DA framework applicable to any DTI prediction model? It would be interesting to see if the benchmarked models also benefit from this.

---

### Official Review · Reviewer_3fJY · 2024-11-02

**Soundness:** 3
**Presentation:** 2
**Contribution:** 2
**Rating:** 3
**Confidence:** 4

**Summary:**

This paper introduces "MoleProLink," a unified framework for unsupervised domain adaptation in drug-target interaction (DTI) prediction, leveraging a blend of advanced mathematical concepts. It proposes the novel DTI-Wasserstein distance metric, which integrates both chemical and structural similarities of drugs and proteins, along with an information-geometric framework that uses the Fisher-Rao metric to map optimal domain adaptation paths. The framework is empirically validated, demonstrating improvements over existing methods in predicting drug-protein interactions across several benchmark datasets.

**Strengths:**

- The paper introduces an innovative approach to DTI prediction by combining concepts from measure theory, information geometry, and optimal transport.
- The framework’s superiority is demonstrated across multiple well-known benchmark datasets (Human, C. elegans, Davis), with strong performance metrics (AUC and AUPRC).

**Weaknesses:**

- Although the paper conducted experiments on well-known benchmarks, these datasets have already reached near-saturation performance levels and are relatively small in size. It would be beneficial if the authors could explain their rationale for choosing these specific datasets and discuss whether they considered using larger, more challenging datasets, such as PDBbind, BindingDB, and KIBA. This would provide valuable insight into their dataset selection process and the potential for expanding their evaluation.
- In addition to the methods compared in the paper, comparison with more recent state-of-the-art models [1,2,3,4] is recommended. Could the authors explain their rationale for choosing the current baselines and why recent models were excluded from the experimental design? Additionally, it would be beneficial to compare the performance of the latest models on the same dataset; if a direct comparison is challenging, a discussion on how these recent models might better predict DTI based on model structure would be valuable.
- While the theoretical framework is well-defined, the paper could benefit from more practical insights into the model's implementation, such as an analysis of the model's time complexity and memory requirements. Additionally, a discussion on how the proposed model can be scaled for large and complex datasets would provide valuable insights into its applicability beyond controlled benchmarks.
- Although benchmark performance is discussed, the model's performance in terms of interpretability and its utility in real-world applications are not fully addressed, which may limit the paper's impact on practical DTI applications that require more than predictive accuracy. It would be beneficial to include a case study analyzing which specific characteristics of DTI the model captures to improve predictive performance, providing further insights into its practical applicability and interpretability.

[1] Zhang, Zuolong, et al. "Enhancing generalizability and performance in drug–target interaction identification by integrating pharmacophore and pre-trained models." Bioinformatics 40.Supplement_1 (2024): i539-i547.

[2] Ahmed, Khandakar Tanvir, Md Istiaq Ansari, and Wei Zhang. "DTI-LM: language model powered drug–target interaction prediction." Bioinformatics 40.9 (2024): btae533.

[3] He, Haohuai, Guanxing Chen, and Calvin Yu-Chian Chen. "NHGNN-DTA: a node-adaptive hybrid graph neural network for interpretable drug–target binding affinity prediction." Bioinformatics 39.6 (2023): btad355.

[4] Zhang, Qi, et al. "FMCA-DTI: A Fragment-oriented method based on a Multihead Cross Attention mechanism to improve Drug-Target Interaction prediction." Bioinformatics (2024): btae347.

**Questions:**

Please see the weaknesses.

---

### Official Review · Reviewer_mQm5 · 2024-11-03

**Soundness:** 2
**Presentation:** 2
**Contribution:** 2
**Rating:** 3
**Confidence:** 3

**Summary:**

The authors proposed a novel mathematical framework, termed MoleProLink) based on several mathematical theories to develop a novel Drug-Target Interaction prediction model. The authors analyze the proposed model over three datasets and three species, showing performance improved or on par with other methods and baseline models.

**Strengths:**

The authors address an important problem in drug repurposing and provides a strong mathematical framework for the newly proposed method. The proposed model also make use of both the protein and drug sequence to predict novel drug-target interactions. The proposed mechanisms are interesting and add over the current existing state-of-the-art.

**Weaknesses:**

The paper is a bit confusing. While the mathematical framework is deeply developed, the description of the experiments lacks important details. Additionally, there are confusing statements over the whole paper (see questions below).

The work uses overly enthusiastic words for very minors improvements, this should be thoroughly revised. For example: "The results presented in Table 2 indicate that our approach performs exceptionally well across all datasets," where the exceptionally well refer to improvements of 0.28% or 3.3%... Also, I guess the authors meant Figure 2 rather than Table 2 as there is no Table 2....

The work lacks a thorough state-of-the-art review, and it performs similarly to the random forest baseline. In recent DTI papers it has been shown that they clearly improve upon RF, so this work should compare against such works (Moltrans, GeNNius, HyperAttanetioDTI, etc.).  These works has shown to perform very well on uncovering novel DTIs from large datasets. Authors should consider performing these analysis, as without this assessment one cannot evaluate the improvements brought by the proposed DTI mathematical framework.

**Questions:**

Are there two or three datasets? In some parts of the paper it is mentioned to work with two datasets and in others with three datasets and three species. Section 3.2 mentions two datasets (Human and DrugBANK), while section 3.4 mentions three datasets on three biological contexts (human, c. elegant, Davis).

---

### Meta-Review · Area_Chair_21uR · 2024-12-20

**Metareview:**

This paper proposes MoleProLink, a framework for drug-target interaction (DTI) prediction, leveraging advanced mathematical theories such as measure theory, information geometry, and optimal transport. The paper demonstrates the performance over the baseline models with sufficient mathematical framework. However, as raised by the reviewers, the clarity and the state-of-the-art baselines are insufficient.

**Additional Comments On Reviewer Discussion:**

Although the paper has some merits, such as providing a strong mathematical framework and demonstrating performance improvements over baseline models, the issues raised by the reviews are critical. For instance, the lack of clarity in the description of experiments and model details (mQm5), the need for a more thorough state-of-the-art review and comparison with recent models (3fJY), and the absence of a clear connection between the theoretical formulation and practical implementation (D6MZ). Additionally, the paper's experimental settings are too simple, and there is a lack of model interpretation and biological insights (Wsa4). There is no authors' response. The decision is reject.

---

### Decision · Program_Chairs · 2025-01-22

Reject